# Using the attachment network Q-sort for profiling one's attachment style with different attachment-figures

Astrid M. Kamperman[1,2‡], Cornelis G. Kooiman[1,3‡*], Nicolas Lorenzini[4], Jurate Aleknaviciute[1], Jon G. Allen[5], Peter Fonagy[4]

1 Department of Psychiatry, Erasmus Medical Center, Rotterdam, The Netherlands, 2 Epidemiological and Social Psychiatric Research Institute, Rotterdam, The Netherlands, 3 Viersprong Institute for Studies on Personality Disorder (VISPD), Halsteren, The Netherlands, 4 Research Department of Clinical, Educational and Health Psychology, University College London, London, United Kingdom, 5 Baylor College of Medicine, Houston, Texas, United States of America

‡ These authors share first authorship on this work.
* kees.kooiman@deviersprong.nl, kooiman@cgkooiman.nl

## Abstract

Attachment instruments vary substantially in practicability of administration, employment of categorical versus dimensional scoring, quality of scales, and applicability to different attachment figures. The Attachment Network Q-sort (ANQ) is a self-report, quasi-qualitative instrument that discriminates relationship-specific attachment styles for multiple attachment figures. The current study assesses the properties of the ANQ in psychotherapy patients and in non-patient respondents, using mother, father and romantic partner as possible attachment figures. Analyzing the ANQ-data with latent class analysis, we found four types or classes of participants: a group with an overall secure profile, a group only insecure for father, a group only insecure for mother, and a group insecure for mother as well as father but not for partner (if available). These profiles proved to have good concurrent, discriminant and construct validity. We conclude that the ANQ is potentially a useful alternative clinical self-report instrument to assess combinations of attachment styles for a range of attachment figures such as parents and a romantic partner.

## Introduction

Attachment theory, as first described by John Bowlby [1–3], is a biopsychosocial model referring to a person's characteristic ways of relating in emotionally important relationships. These ways of relating are initialliy learned during early infancy and mold subsequent intimate relationships. Adults who are securely attached have internalized a reliable relationship with his/her caregivers in infancy, with stable and nuanced self-other representations, good mentalizing capacities and thus an adequate equilibrium between self-regulation and interpersonal regulation of stress. Insecurely attached individuals lack these capacities and they are prone to use inadequate strategies to cope with stressful events, eventually leading to emotional

**Data Availability Statement:** Data are stored at the institutional database of the Erasmus Medical Centre in Rotterdam, The Netherlands. The datasets on which the analyses are based are available on

request to the Local Ethics Committee of the Erasmus Medical Centre in Rotterdam, due to ethical restrictions and patient confidentiality requirements. To request the data, please contact: Dr Astrid Kamperman: a.kamperman@erasmusmc.nl or Dr Joke Tulen: j.h.m.tulen@erasmusmc.nl.

**Funding:** CGK received a grant from the Dutch Psychoanalytic Funds (pafondsen@outlook.com; no website available). The funder had no role in study design, data collection and analysis, decision to publish, or preparation of the manuscript.

**Competing interests:** The authors have declared that no competing interests exist.

dysregulation. Insecure attachment is indeed associated with personality disorder, and with mood and anxiety disorders [4–8].

Because of its heuristic quality, attachment theory became appealing to clinicians of diverse psychotherapy orientations as well as to researchers in the clinical and psychobiological domains [6,9,10]. Various instruments, with diverse strengths and weaknesses, have been developed to support research and/or clinical practice. To meet theoretical and practical issues, that will be outlined below, Fonagy and colleagues [11–13] added the Attachment Network Q-sort (ANQ) to the armamentarium to assess attachment styles. This article is focused on the psychometric properties of the ANQ, starting with a discussion of other attachment instruments in order to clarify the need for yet another measure.

Mary Ainsworth and colleagues [14,15] were the first to develop an instrument to assess attachment. They constructed a laboratory test, the Strange Situation Procedure (SSP), to observe and code the attachment behavior of toddlers upon separation and reunion with the parent, which could be categorized as secure, avoidant or anxious. Later Main & Solomon [16] added an extra category 'disorganized' for children who could not be classified exclusively within one of the aforementioned categories. The SSP is a laboratory test appropriate for children up to four years of age, as separation of the parent is less stressful when children get older.

Subsequently, Main and colleagues [17] developed the Adult Attachment Interview (AAI) which made it possible to study the impact of the parents attachment style on the development of their offspring's attachment style as measured with the SSP. The AAI is a semi-structured interview that can be applied in the consulting room. Respondents are asked open-ended questions about the attachment relationships with their parents during childhood and about the influence of these relationships on their own development. The answers of the respondents are documented verbatim and coded on different scales, the most important being the 'coherence-scale'. In this way, the attachment classification towards the parents is indirectly inferred by linguistic cues, and by the overall coherence and believability of the respondent's narrative. Respondents are classified as secure, dismissing, preoccupied, unclassifiable, and/or unresolved with regard to traumatic experiences. The development of the AAI fostered research on adult attachment and its associations with personality, parenting and pathology [9]. The AAI is also appealing for clinicians as it generates a wealth of biographical and emotional material. However, the AAI has major practical drawbacks, as it is time-consuming and complicated to score, requiring extensive training [18]. These factors hinder the use of the AAI in large scale research as well as the implementation of the instrument in regular psychotherapeutic practice.

A different line of research has been developed in the field of personality and social psychology. Hazan and Shaver [19] developed a brief categorical self-report measure of adult attachment which requires respondents to characterize themselves according to three short vignettes representing a secure, avoidant and anxious attachment style in romantic relationships. Subsequently, numerous self-report instruments to assess attachment have been developed, many of them multi-item, Likert-scale instruments that assess attachment styles dimensionally, like the Attachment Style Questionnaire (ASQ) [20], the Relationship Questionnaire (RSQ) [21], the Adult Attachment Questionnaire (AAQ) [22], the Adult Attachment Scale (AAS) [23] and others, among which the Experiences in Close Relationships (ECR) [24] and ECR-revised [25,26] are considered to have the best psychometric properties [27,28]. These multi-item, self-report instruments probe for conscious attitudes, feelings and thoughts concerning actual 'close relationships' in this way assessing attachment towards 'a romantic partner' or towards 'close relations in general'.

Although easy to use, these self-report instruments have downsides too. Self-report questionnaires as the ECR(-r) are sometimes vague and variable with regard to the potential attachment figure targeted, for example a specific 'romantic partner' or 'how one generally feels in close relationships'. The Likert scales are liable to facilitate response bias by halo effects and it has been questioned whether conventional self-report questionnaires can be used to assess personality profiles [29,30].

Other differences among assessment strategies merit discussion. One of these is whether attachment should be measured categorically or dimensionally, although taxometric analyses of available data seem to support the dimensional option even for the AAI, despite being originally developed as a categorical instrument [31,32]. Another topic of discussion is whether one has one dominant and generalized attachment style versus relationship-specific attachment patterns. Theoretically, it can be expected that different attachment styles can be activated in different relationships, as one may have been treated differently by diverse early caregivers [11,27,33–35]. However, as already observed by Collins & Read in 1994 [36], there is a general tendency to discuss attachment as a single and simple character trait. Yet, as early as 1981, Main and Weston [37] reported that in a study with the SSP some toddlers exhibit different attachment styles towards their mother and father. This finding has been replicated by other researchers later on [38]. Furthermore, using the Relationship Questionnaire, Ross & Spinner [39] found that adults also report different attachment style profiles dependent on the specific attachment figure they refer to. Apart from having possible different attachment profiles towards different potential attachment figures, Crittenden [34] emphasizes that relationships also have non-attachment qualities that, depending on the relationship, might be more important than the attachment qualities. This makes it possible that, in the construction of assessment instruments, the endorsement of items reflecting secure attachment might be confounded with affectively positive non-attachment experiences of a relationship (e.g. liking) and the endorsement of items reflecting insecure attachment with items reflecting non-attachment negative affective experiences of a relationship (e.g. disliking) [11].

In an attempt to counter such problems, Fonagy and colleagues [11–13] developed a new instrument to assess adult attachment, the Attachment Network Q-sort (ANQ). As an alternative for the conventional self-report questionnaires with their described drawbacks, the Q-sort technique was used in the development of the ANQ [40]. In the domain of attachment research also some other Q-sort instruments have been developed, for example, the Attachment Q-sort [41] that assesses attachment characteristics of children in their natural environment, the Q-sort scoring and analyses of the AAI [42], and the California Adult Q-sort that assesses adult romantic attachment orientation [43]. However, these instruments are observer-scored, and the ANQ is intended to be an easily administered self-report instrument that additionally probes potentially different attachment styles with different attachment figures while discriminating between attachment qualities and non-attachment affective valences of relationships.

The Q-sort methodology [30] consists of a Q-sorting procedure followed by a Q-pattern analysis. In the Q-sorting procedure respondents are asked to assign a number of randomly presented items a ranking position in a fixed quasi-normal distribution along a simple, face-valid distribution (e.g. most characteristic to most uncharacteristic) [44]. Each ranking position or pile has a limited number of items that can be assigned to it. With this 'forced' distribution respondents are forced to weigh the importance of items relative to each other. The Q-sort methodology aims to make gestalt configurations of the items typical for a respondent, as well as a clustering of persons with similar profiles. As such, Q-sort tests are considered semi-qualitative or quanti-qualitative instruments [30].

Building upon items from existing attachment instruments, Fonagy and colleagues [11–13] started with 226 items of which 136 items were hypothesized to be attachment items and 90 items to be non-attachment (affectively valenced) items. After empirical evaluation for consensus by a group of international experts in the field of attachment, 60 items were selected on the basis of the highest agreement, adequate internal consistency and test-retest reliability. The items were considered balanced for attachment and non-attachment qualities of relationships and for social desirability, and they were designed for computerized self-administration. The number of items are as follows: secure (n = 20 items), dismissing or avoidant (n = 10 items), preoccupied (n = 10 items), and 10 items each for respectively positively and negatively valenced relationship descriptors that are not specific to attachment relationships (so called non-attachment items) (see S1 Appendix ANQ-items in S1 File). A computer program was developed to administer the ANQ with the possibility to assess current attachment qualities for any number of potential attachment figures like parents, romantic partner or psychotherapist [13]. With each attachment figure, items are presented randomly to the respondent who is asked to rank the items in seven categories: mostly untrue (3 stacks), quite untrue (6 stacks), slightly untrue (12 stacks), mixed (18 stacks), slightly true (12 stacks), quite true (6 stacks) and mostly true (3 stacks).

In this study, we first explored whether the ANQ is capable of assessing different attachment styles, and whether we could distinguish distinct homogeneous classes or subgroups of participants with similar attachment-style profiles with regard to three different potential attachment figures: mother, father and romantic partner. We subsequently explored the concurrent validity of the ANQ with the ECR-r. Next, we studied the clinical relevance of these subgroups by relating them to relationship affective valence, current symptomatology, various dimensions of personality pathology, and a history of abuse and/or neglect. Finally, we examined the added value of this new instrument, by testing the performance of the ANQ in predicting caseness in comparison to the ECR-r. We hypothesize that respondents with an insecure attachment style towards all key-figures suffer from several indices for psychopathology more frequently and severely than respondents who have a secure attachment style towards one or more key-figures.

## Material and methods

### Participants

The participants in this study stem from two separate samples that for the purpose of this study were taken together. The full sample consists of 510 participants.

The English sample was a convenience sample of men and women out of the general London population. Recruitment was made from the community by advertisement in newspapers as well as by posters. Participants were paid standard rates for taking part in psychological tests. Inclusion was by age [18–65] and language competence. No additional in- or exclusion criteria were formulated except sufficient competence in the English language to permit participation. Participation in the survey was voluntary. Permission for conducting this part of the study was obtained from the University Ethics Committee of University College London.

The Dutch sample consisted of female psychotherapy outpatients of reproductive age and healthy females, matched by age, from the general population who were recruited through posted flyers and local internet advertisements. Participants from both groups were inhabitants of the Rotterdam municipal area. As the Dutch respondents participated in a larger study on the psychophysiological responsivity to psychological stress [45], all participants (patients and healthy respondents) underwent the Structural Clinical Interview for DSM-IV axis I disorders (SCID-I) (by JA). Patients were considered ineligible to participate if they had comorbid

diagnoses of bipolar disorder, schizophrenia, current mood disorder, or the use of psychotropic medication within the previous nine months. Eligibility requirements for the healthy participants included absence of any DSM-IV axis I, and no history of psychiatric or psychological treatment. All Dutch subjects underwent a somatic screening prior to study enrollment. Somatic exclusion criteria included: a) a history of any neurological or endocrine disorders, b) drug or alcohol abuse within the previous four months, c) BMI < 18 or BMI > 30, d) current pregnancy or lactation. All the participants had fluent command of Dutch language. Written informed consent was obtained from all participants. This part of the study was approved by the Medical Ethical Research Committee of the Erasmus MC, University Medical Center Rotterdam.

## Instruments

*The Attachment Network Q-sort* (ANQ-sort) [11–13] has been described in the introduction of this manuscript. The translation of the ANQ-sort from English into Dutch took place according to the International Test Commission guidelines [46]. For a description of the scoring of the ANQ we refer to the final scoring tool that is added as a supplement (S5). In this study current attachment style was assessed with mother, father and romantic partner (if available) as key figures. Respondents were asked to characterise their feelings towards these key figures in terms of the ANQ items. The completion time for the three key figures was about forty minutes. The evaluation of the first figure takes a bit longer than the following two attachment figures, as people needed time to read and comprehend the items and the procedure the first time. Most participants (84.5%; 431/510) completed the ANQ with regard to all three presented key figures: their mother, father and romantic partner. Six participants completed the ANQ for one key figure only (1.2%; 6/510). In two of these cases, only attachment to mother was assessed; one case only assessed attachment to father, and in three cases only attachment to the romantic partner was assessed. The remaining participants completed the ANQ for two key figures (14.3%; 73/510), the vast majority of them for mother and father. This resulted in 1445 completed ANQ questionnaires, representing attachment to mother (n = 503), father (n = 499), and romantic partner (n = 443).

*The Experiences in Close Relationships-revised* (ECR-r) [25,26,47] is a self-report questionnaire with two reliable 18-item subscales for the dimensional assessment of attachment-related anxiety and attachment-related avoidance. Low scores on both dimensions are considered to indicate attachment security. In accordance with the common instruction of the instrument, participants were asked to think about their romantic partner while rating the appropriateness of each item on a 7-point Likert scale, whereas participants without a current partner were asked to rate how they felt generally during intimate relationships.

*The Brief Symptom Inventory* (BSI) [48,49] is a commonly used self-report questionnaire with 53 items on a five-point Likert scale about general psychiatric complaints and symptoms during the past two weeks. The total score gives a general measure for the severity of psychopathology. The mean total score and scales have a theoretical range of 0–4 with higher scores meaning greater pathology. According to de Beurs & Zitman [49] a score of 0.84 or higher indicates the presence of a psychiatric disorder.

*The Dimensional Assessment of Personality Pathology short form* (DAPPsf) [50,51] is the abbreviated version of the DAPP-BQ. The DAPPsf has 136 items with a five-point Likert scale assessing DSM-IV personality pathology. Scales have a theoretical range of 1–5 with higher scores meaning greater pathology. The scales cover the domains emotional dysregulation, dissocial behavior, inhibition, compulsivity and self-harm, and have adequate internal

consistencies. A cut-off ≥ 3.1 mean score on the scale for Identity Problems has been established as an index for the presence of personality disorder [52].

*The Inventory of Interpersonal Problems* (IIP-64) [53–55] is a shortened version of the 127-item original. The IIP-64 is a clinically useful instrument in the domain of personality functioning and psychotherapy as it assesses dysfunctional attitudes in interpersonal encounters. The IIP-64 has eight scales theoretically grouped along the dimensions dominance and affiliation, and they are known to have adequate internal consistencies. Items are scored on a 5-point Likert scale.

*The Positive and Negative Affect Scale* (PANAS) [56] has two reliable scales with 10 items each, measuring positive (PA; e.g. energetic, inspired) and negative affectivity (NA; e.g. angry, upset). The items are scored on a 5-point Likert scale ranging from 1–5. The PANAS is designed to measure affect in various contexts such as 'at present' or 'in general'. We used the time-frame 'in general' to reflect dispositional affect.

*The Childhood Trauma Questionnaire* (CTQ-sf) [57–59] is a 28-item self-report questionnaire to assess childhood abuse and neglect through five scales with adequate to good internal consistencies: physical, sexual and emotional abuse and physical and emotional neglect. Items are rated on a five-point Likert scale.

The ANQ-sort and the BSI were used in the English as well as in the Dutch samples. The other instruments were used in the Dutch sample only.

## Statistical methods

To examine the existence of homogeneous subgroups of adults based on the attachment to their mother, father and romantic partner, we employed a two-tier approach to our analyses. First, we tested the structural validity of the ANQ questionnaire by testing the fit of the data to the theoretical factor structure model of attachment [12]. We used confirmatory factor analysis (CFA) to analyze the data of the English sample. The factor structure was tested using mother, father, and romantic partner separately as attachment figures. These analyses were repeated for the Dutch healthy control and patient samples. Then, we tested the invariance of the factor structure over the attachment figures, using multi-group confirmatory factor analysis. Next, we tested the invariance over the attachment figures in the full sample. Finally, we tested the invariance of the factor structure over the English sample, Dutch control sample and Dutch patient sample. This first tier of analyses aims to confirm the theoretical structure of the ANQ, and to explore the structural invariance of the ANQ over attachment figures and populations. For the analyses of our second tier we used latent class analysis using the full sample to distinguish homogeneous subgroups of participants, based on the specificities of the attachment to their mother, father and partner. We determined the number of classes based on the goodness of fit of the model, in addition to theoretical and clinical interpretability, and parsimonious criterion. Finally, we tested the concurrent and discriminant validity of the classes of participants based on attachment profile. For this aim we examined the association of the attachment classes to demographic variables and to psychiatric symptomatology, interpersonal problems and other aspects of personality pathology in the Dutch sample. In the English sample discriminant validity was tested only with regard to psychiatric symptomatology assessed using the BSI.

CFA analyses were conducted using robust maximum likelihood estimation (RML). Although the impact of the Q-sort methodology on conventional multivariate analysis and multivariate normality is unknown, we assumed RML to result in sufficiently robust findings, since the underlying ANQ-items have more than 5 ordinally ordered response categories [60]. All forty attachment items were included in the analysis (ANQ items 1–40, see S1 Appendix in S1 File). The factor structure was adapted using modification indices (>10.0), R-squared

(<0.10), and (negative) residual variances. Modifications were only performed if they were theoretically justifiable and did not influence the estimates of other parameters in the model. The fit of the models was evaluated using theoretical judgement on the interpretability of the factors and statistical goodness-of-fit indices, e.g. Comparative Fit Index (CFI), Tucker-Lewis Index (TLI), Root Mean Square Error of Approximation (RMSEA) and Standardized Root Mean Square Residual (SRMR). Fit is considered acceptable in case of a Chi2/df ratio < 3.0; RMSEA < 0.08, CFI ≥ 0.90, TLI ≥ 0.95, and SRMR <0.08 [61–64] (see S2 Appendix Confirmatory Factor Analyses in S1 File).

The thus confirmed factor structure for attachment to mother was then tested for invariance using Multi-Group Confirmatory Factor Analysis (MGCFA). In line with common practice in psychological research, we chose to test using linear MGCFA [61,65]. Similar to our CFA analyses, linear MGCFA was conducted using robust maximum likelihood estimation [60]. We started with the specification of a configural invariance model, using the three-factor model. Next we evaluated the metric and scalar invariance, i.e. factor loadings and intercepts were assumed invariant over the groups. Variances and covariances were allowed to differ. Fit of the nested models was described using Chi2, CFI, TLI, RMSEA, and SRMR [61,62]. A MGCFA model was deemed invariant based on the Chi2-tests with Satorra-Bentler correction (p>0.05) [66], and absolute change of the CFI and RMSEA indices, i.e. ΔCFI, ΔRMSEA ≤ 0.02 [65,67]. Analyses were conducted over the two subgroups with acceptable fit, first (e.g. mother and father as attachment figure). Then, repeated over all subgroups (e.g. mother, father, and romantic partner as attachment figure). The results from our MGCFA modelling procedure are reported in S3 Appendix Multi-Group Confirmatory Factor Analyses in S1 File.

Latent Class Analyses (LCA) for continuous variables, often refered to as Latent Profile Analyses, were conducted using robust maximum likelihood estimation (RML). We calculated the mean scores of the factors of the ANQ for the mother, father and romantic partner (e.g. nine sum factor scores) and used them as input for the analysis. The number of extracted profiles ranged between 2 and 7. Several goodness-of-fit indices were used to determine the optimal number of latent profiles and the overall fit of the model, including the loglikelihood-value, Akaike's (AIC) and (adjusted) Bayesian Information (BIC) criteria, entropy, the Lo-Mendell-Rubin adjusted likelihood ratio test (LMR-test), and parametric bootstrapped likelihood ratio-test. The distinctiveness of the class is evaluated using entropy. A higher entropy proportion indicates a clearer distinction between subgroups or classes with different attachment profiles. Values above 80% are desirable [68–73]. The selection of the best model in the LCA was based on these indices and the theoretical interpretation of the distinct profiles. Participants were allocated to a specific class based on their highest posterior class probability.

Association between the obtained attachment subgroups and proximal variables were formally tested using Chi2-tests for categorical variables, ANOVA's for normally distributed continuous variables, and Kruskal-Wallis tests for non-normally distributed continuous data. Correlations between ANQ-subscales were calculated using Pearson's rho test. We refrained from conducting additional (post-hoc) tests between profiles and samples to limit the risk of type I errors.

Performance of the ANQ-sort in predicting caseness, and the added predictive performance relative to the ECR-r was quantified according to the pseudo $R^2$-measures of Cox and Snell [74] and Nagelkerke [75]. Higher $R^2$-values indicate beter predictive performance.

We used SPSS 23.0 for data-management and descriptive analysis. Factor and latent class analyses were performed using MPlus version 7.4 software [76].

## Availability of databases

Data are stored at the institutional database of the Erasmus Medical Centre in Rotterdam, The Netherlands. The datasets on which the analyses are based are available on request to the Local Ethics Committee of the Erasmus Medical Centre in Rotterdam, due to ethical restrictions and patient confidentiality requirements. To request the data, please contact: Dr Astrid Kamperman: a.kamperman@erasmusmc.nl or Dr Joke Tulen: j.h.m.tulen@erasmusmc.nl.

# Results

## Sample descriptions

The English sample was a convenience sample (n = 340) of the general population with 220 females (65%) and 120 (35%) males ranging in age from 18–61 years (M = 36.0, SD = 11.0).

The Dutch sample consisted of 96 female psychotherapy outpatients and 74 healthy females from the general population. Together they ranged in age from 19–50 years (M = 29.5, SD = 7.5). For the Dutch participants we had some additional sociodemographic data. Thirty-seven percent of the Dutch participants were unmarried. Sixty-nine percent were highly educated, 29% had a middle education degree and 2% received only lower education. Regarding their source of income: 52 (31%) were students, 100 (59%) were employed, 12 (7%) were unemployed, 4 (2%) were receivers of sickness benefit and one participant (0.6%) was a homemaker.

Of the 96 participating patients, 44% had one or more DSM-IV axis I diagnoses: 22% an anxiety disorder, 10% an obsessive compulsive disorder, 19% eating disorder, and 10% post-traumatic stress disorder. No patients with a mood disorder participated as this was an exclusion criterion. Eighty-four percent of the patients had one or more DSM-IV axis II diagnoses: 40% cluster B, and 67% cluster C.

## Results from CFA and MGCFA

The CFA showed an adequate fit of the theoretical model, including three separate factors in the English sample assessing attachment to mother. Best fit was obtained by removing 11 items (Chi2 = 962, $p <$ .001; RMSEA = .068, 95%CI: .063 to .073; CFI = .92; TLI = .91; SRMR = 0.59) in the English sample (S2 Appendix, Table 1a in S1 File). Comparable fit was obtained in assessing attachment to father (Chi2 = 817, $p <$ .001, RMSEA = .059 (95%CI: .054 to .065), CFI = .95, TFI = .94, SRMR = 0.052) (S2 Appendix Table 1b in S1 File). Fit of the theoretical model in the data regarding attachment to romantic partner was considerably less good (Chi2 = 1021, $p <$ 0.001, RMSEA = .071 (95%CI: .066 to .077), CFI = .83, TFI = .81, SRMR = 0.069) (S2 Appendix Table 1c in S1 File). Multigroup CFA showed acceptable configural and metric invariance over mother and father as attachment figures in the English sample (Chi2(26) = 23; $p$ = 0.63; ΔCFI = 0.001; ΔRMSEA = 0.002) and in the full sample (Chi2(26) = 32; $p$ = 0.19; ΔCFI = 0.005; ΔRMSEA = 0.001) (S3 Appendix Tables 2a-2c in S1 File). However, regarding the romantic partner as attachment figure, configural was less optimal as expected considering the results from the CFA analysis (Chi2 = 2539, $p <$ 0.001, RMSEA = .061 (95% CI: .058 to .064), CFI = .83, TLI = .81, SRMR = 0.065) (S3 Appendix Tables 2d-2f in S1 File). The first factor 'Secure Attachment' consisted of 16 items. The second factor, 'Dismissing attachment' was formed by 8 items. The third factor 'Preoccupied attachment' was described by 5 items. (see S1 Appendix in S1 File for the instrument and the items retained in the analyses). The full CFA and MGCFA modelling procedures are reported in S2 and S3 Appendices in S1 File.

## Results from LCA

The results of the LCA show that model fit in terms of AIC, BIC and adjusted BIC increases with increase of classes (Table 1). Based on the LMR-test, a model describing four separate homogeneous subgroups of participants shows better statistical fit than a model with three classes. A model with five classes did not show a significantly better fit over a model with four classes as shown by the LMR-test. Entropy dropped from 91% to 90%, indicating a worsening of fit of the five-class model. However, a model containing six classes showed significantly better fit than the five-class model, in combination with an increase of entropy. The parsimony criterion, however, states that a model with fewer parameters is preferred [62]. Therefore, we concluded that a model describing four classes shows the best fit based on statistical fit, theoretical interpretation and parsimony.

Fig 1 shows the attachment styles of the four different subgroups with well discernable profiles. We labelled these groups: overall secure (OS), insecure for father (IF), insecure for mother (IM), and insecure for both parents (IFM). More than half of the participants (n = 263; 52%) cluster in a subgroup showing a stable pattern of high scores on secure attachment to mother, father and romantic partner, in combination with low scores for these attachment figures on dismissing and preoccupied attachment (OS). Almost a quarter of the participants (n = 124; 24%) depict an insecure level of attachment to their father, with high scores on dismissing and preoccupied attachment while their attachment to mother and romantic partner can be described as secure (IF). Fifteen percent of participants (n = 80) depict an attachment style characterized by an insecure, dismissing and preoccupied, attachment to mother, and secure attachment to father and romantic partner (IM). Finally, fewer than 10% of the participants (n = 43) displayed an attachment style characterized by insecure, dismissing and preoccupied, attachment to both their mother and father while their attachment to their romantic partner could be described as secure, with low scores on dismissing and preoccupied attachment (IFM). There were no subgroups with participants who reported their experienced attachment to their romantic partner as insecure.

**Table 1. Latent class modelling procedure and fit.** Analyses are conducted using the full sample (N = 510).

| | Number of latent classes included in the model | | | | | |
|---|---|---|---|---|---|---|
| | 2 | 3 | 4 | 5 | 6 | 7 |
| Loglikelihood | -4565 | -4311 | -4145 | -4029 | -3948 | -3894 |
| AIC | 9185 | 8697 | 8386 | 8173 | 8032 | 7943 |
| BIC | 9304 | 8858 | 8589 | 8419 | 8320 | 8273 |
| Adjusted BIC | 9215 | 8737 | 8437 | 8235 | 8104 | 8026 |
| Entropy | 91% | 90% | 91% | 90% | 91% | 90% |
| Lo-Mendell-Rubin adjusted LR-test* | 2 vs 1<br>957<br>P<0.00001 | 3 vs 2<br>500<br>P = 0.0071 | 4 vs 3<br>326<br>P = 0.0093 | 5 vs 4<br>229<br>P = 0.2361 | 6 vs 5<br>159<br>P = 0.0485 | 7 vs 6<br>97<br>P = 0.4879 |
| Parametric Bootstrapped LR-test* | P<0.00001 | P<0.00001 | P<0.00001 | P<0.00001 | P<0.00001 | P<0.00001 |
| N for each class | 1: N = 336 (66%)<br>2: N = 174 (34%) | 1: N = 99 (19%)<br>2: N = 285 (56%)<br>3: N = 126 (25%) | 1: N = 124 (24%)<br>2: N = 80 (15%)<br>3: N = 263 (52%)<br>4: N = 43 (9%) | 1: N = 34 (7%)<br>2: N = 125 (24%)<br>3: N = 78 (15%)<br>4: N = 230 (45%)<br>5: N = 43 (8%) | 1: N = 37 (7%)<br>2: N = 14 (3%)<br>3: N = 221 (43%)<br>4: N = 34 (7%)<br>5: N = 126 (25%)<br>6: N = 78 (15%) | 1: N = 83 (16%)<br>2: N = 183 (36%)<br>3: N = 27 (5%)<br>4: N = 98 (19%)<br>5: N = 58 (11%)<br>6: N = 33 (6%)<br>7: N = 28 (5%) |

* A significant result indicates that a model with k classes fits better than a model with k-1 classes

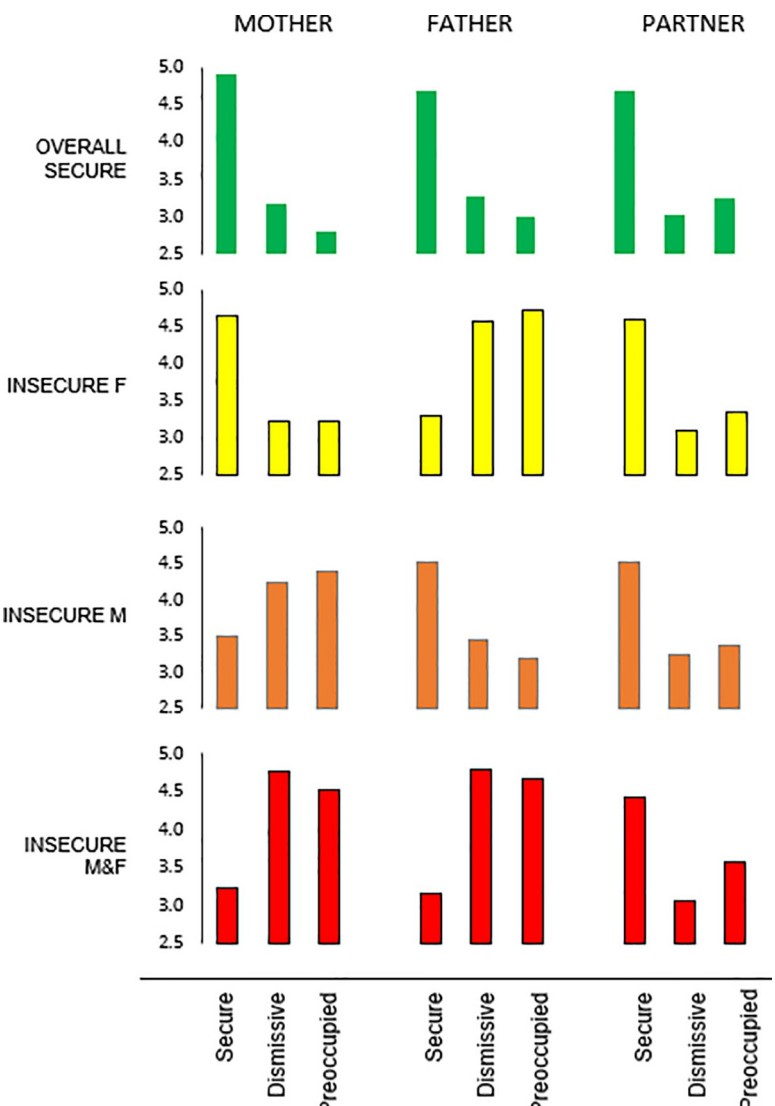

**Fig 1. The four profile groups of participants regarding their potential attachment figures.**

## The association of the attachment profile groups with demographic characteristics

The healthy control group has the highest prevalence (74%) of overall secure participants and the lowest prevalences of participants in all the insecure subgroups. The patient group has the lowest prevalence (33%) of overall secure participants and the highest prevalences of participants in all insecure subgroups. The participants of the general population scored in between. No gender differences were found. The participants of the four attachment subgroups differed with respect to age, participants with the overall secure profile being somewhat younger than participants of the other groups. From the data of the Dutch sample, it can be seen that overall the participants with the IFM-profile do poorer socially than the other groups, especially with respect to the group with the overall secure profile. They have lower educational levels, their employment status is less beneficial and they tend to have more frequent smoking and drug use habits (see Table 2).

**Table 2. Associations with demographic characteristics.**

|  | Overall Secure N (%) N = 263 | IF N (%) N = 124 | IM N (%) N = 80 | IFM N (%) N = 43 | Test |
|---|---|---|---|---|---|
| Sample* |  |  |  |  | Chi2(3) = 31.30; p<0.001 |
| General population (UK) | 176 (51.8) | 80 (23.5) | 53 (15.6) | 31 (9.1) |  |
| Healthy control (NL) | 55 (74.3) | 14 (18.9) | 4 (5.4) | 1 (1.4) |  |
| Mentally ill (NL) | 32 (33.3) | 30 (31.3) | 23 (24.0) | 11 (11.5) |  |
| Gender |  |  |  |  | Chi2(3) = 0.92; p = 0.820 |
| Male | 65 (22.6) | 28 (22.6) | 16 (20.0) | 11 (25.6) |  |
| Female | 198 (75.3) | 96 (77.4) | 64 (80.0) | 32 (74.4) |  |
| Age (mean; SD) | 32.5 (10.4) | 34.5 (9.9) | 35.5 (10.2) | 37.1 (11.3) | F(3) = 3.54; p = 0.015 |
| Committed relationship (Abs; % Yes) | 56 (65.1) | 26 (59.1) | 15 (55.6) | 9 (75.0) | Chi2(3) = 1.83; p = 0.609 |
| Educational level |  |  |  |  | Chi2(12) = 23.58; p = 0.023 |
| Lower vocational | 1 (1.2) | 1 (2.3) | 0 (-) | 1 (8.3) |  |
| Middle vocational | 14 (16.3) | 14 (31.8) | 11 (40.8) | 8 (66.7) |  |
| Preparatory academic | 1 (1.2) | 1 (2.3) | 0 (-) | 0 (-) |  |
| Higher vocational | 37 (42.9) | 16 (36.3) | 7 (25.9) | 2 (16.7) |  |
| Academic | 33 (38.4) | 12 (27.3) | 9 (33.3) | 1 (8.3) |  |
| Employment |  |  |  |  | Chi2(12) = 25.19; p = 0.014 |
| Student | 34 (39.1) | 9 (20.5) | 8 (30.8) | 1 (8.3) |  |
| Employed | 48 (55.2) | 30 (68.1) | 14 (53.8) | 8 (66.7) |  |
| Unemployed | 5 (5.7) | 4 (9.1) | 2 (7.7) | 1 (8.3) |  |
| Sickness benefit | 0 (-) | 1 (2.3) | 1 (3.8) | 2 (16.7) |  |
| Homemaker | 0 (-) | 0 (-) | 1 (3.8) | 0 (-) |  |
| Smoking (absolute;% Yes) | 22 (26.2) | 4 (9.3) | 7 (29.2) | 5 (45.5) | Chi2(3) = 8.55; p = 0.036 |
| Alcohol taking (absolute; % Yes) | 54 (64.3) | 22 (51.2) | 14 (60.9) | 8 (72.7) | Chi2(3) = 2.76; p = 0.430 |
| Drug taking (absolute; % Yes) | 2 (2.4) | 1 (2.3) | 3 (13.0) | 2 (18.2) | Chi2(3) = 9.07; p = 0.028 |

Apart from Age and Gender, no information available from London sample. All other comparisons are calculated using the Dutch sample only

* Percentages calculated by row

Overall Secure: a secure attachment style regarding mother, father and romantic partner; IF: secure attachment style regarding mother and romantic partner, but not father; IM: secure attachment style regarding father and romantic partner, but not mother; IFM: insecure attachment style towards father as well as mother, but not partner.

### Concurrent validity with the ECR-r

Differences between the attachment classes with respect to ECR-r scores were assessed using ANOVA. No post-hoc tests were performed. Attachment class membership and attachment style as assessed using the ECR-r showed significant associations. Participants with the overall secure profile demonstrated the lowest scores on the attachment related anxiety and avoidance subscales of the ECR-r. Incrementally higher scores on the ECR-r subscales were found for participants of the IF-class, the IM-class, and the IFM-class, respectively (Table 3).

### Construct validity with affective valence of relationships, symptomatology, personality features, and history of abuse and neglect

Differences between the attachment classes with respect to affective valence, symptomatology, personality and history of abuse and neglect were assessed using ANOVA and Kruskal-Wallis tests. Overall, insecure attachment to mother or father was associated with more negative affective valence in relationships with these parents (Table 4). Participants of the IFM-class had the most negative affective relationships with their mother and father.

**Table 3. ANQ-sort and concurrent validity.**

| | Overall Secure N = 83 | IF N = 42 | IM N = 24 | IFM N = 12 | test |
|---|---|---|---|---|---|
| ECR–r (mean; SD)* | | | | | |
| attachment related avoidance | 2.4 (0.9) | 3.2 (1.0) | 3.3 (1.2) | 3.5 (1.2) | F(3) = 9.74; p<0.001 |
| attachment related anxiety | 2.8 (1.3) | 3.8 (1.3) | 4.2 (1.4) | 4.3 (1.1) | F(3) = 10.97; p<0.001 |

ECR-r = Experiences in Close Relationships revised; Overall Secure: a secure attachment style regarding mother, father and romantic partner; IF: secure attachment style regarding mother and romantic partner, but not father; IM: secure attachment style regarding father and romantic partner, but not mother; IFM: insecure attachment style towards father as well as mother, but not partner.

* ECR-r results is missing in nine cases.

Participants of all but one attachment class membership tended to show more positive than negative affective valences in their relationships. Only when relationships with both parents were insecure did the negative valence outweigh the positive valence. Security with parents was not significantly related to the affective valence of relationships with romantic partners.

Postive affective valence was correlated significantly with the ANQ subscales (*r* ranges from .43 to .51), as was negative affective valence (*r* ranges from .50 to .77).

Attachment-class membership was significantly associated with psychiatric symptomatology as measured with the BSI; attachment-class membership and the BSI subscales all had significant associations with the lowest scores consistently found for the participants with the overall secure profile. Also, the lowest frequency for BSI-caseness was found for participants of the Overall Secure (OS-)class. (Table 4).

With regard to the mood disposition as measured with the PANAS, participants in the OS-class showed the highest levels of positive affectivity and the lowest levels of negative affectivity while participants of the IFM-class showed the lowest levels of positive affectivity and the highest levels of negative affectivity (Table 4).

Participants of the different ANQ classes also scored differently and meaningfully on most of the DAPPsf-scales reflecting personality pathology (Table 4). Participants of the OS-class showed low levels of most pathological personality traits, while participants belonging to the IFM-class generally showed the highest levels of pathological personality traits and the participants of the IF- and IM-classes generally scoring intermediate. Participants belonging to the IF-, IM- and IFM-classes also showed high proportions of scorers above the cut-off on the Identity Problems subscale, indicating the presence of personality disorder. In contrast, only 11% of the participants of the OS-class scored above this cut-off.

Analogous results were found for the associations with interpersonal problems as measured with the IIP-64. The lowest negative traits were consistently shown by participants of the OS-class, the highest scores by the participants of the IFM-class, and the other two classes scored in between (Table 4).

With regard to childhood abuse and neglect, we found that least abuse and neglect was reported by the participants of the overall secure class, while most abuse and neglect was reported by the participants of the IFM-class. Again, participants with insecure attachment to father or insecure attachment to mother showed intermediate levels of abuse and neglect (Table 4).

Table 5 shows the predictive performance of the ANQ-sort relative to the ECR- subscales. Based on the model fit measures, the two ECR-r subscales combined predict BSI- and DAPP-sf-caseness better than the ANQ-sort. Additionally, with regards to BSI-caseness, we only found a small and non-significant improvement of the model when the ANQ-sort was added to the ECR-r model. However, with regards to the DAPP-sf caseness, the addition of the

**Table 4.  ANQ-sort and construct validity.**

| | Overall Secure | IF | IM | IFM | Test |
|---|---|---|---|---|---|
| **APPRAISAL OF RELATIONSHIP (ANQ-sort; mean; sd)** * | | | | | |
| positive non-attachment M | 46.7 (3.5) | 46.0 (5.0) | 41.7 (5.8) | 39.3 (5.8) | F(3;502) = 52.89; p<0.001 |
| negative non-attachment M | 30.8 (3.6) | 32.0 (4.7) | 40.4 (6.4) | 43.3 (6.2) | F(3;502) = 153.08; p<0.001 |
| positive non-attachment P | 49.1 (4.0) | 41.3 (6.1) | 48.4 (3.9) | 42.7 (6.2) | F(3;498) = 85.84; p<0.001 |
| negative non-attachment P | 30.6 (4.2) | 41.4 (7.6) | 30.9 (4.9) | 41.3 (7.0) | F(3;498) = 137.65; p<0.001 |
| positive non-attachment partner | 49.0 (4.2) | 48.8 (4.6) | 48.6 (4.4) | 48.3 (5.4) | F(3;442) = 0.41; p = 0.748 |
| negative non-attachment partner | 31.1 (5.3) | 31.2 (5.2) | 31.9 (6.0) | 33.3 (7.7) | F(3;442) = 1.98; p = 0.117 |
| **BSI (median; IQR)** * | | | | | |
| Somatic Complaints | .29 (.00-.57) | .43 (.14-.86) | .50 (.00–1.14) | .29 (.00–.71) | KW(3) = 11.58; p = 0.009 |
| Cognitive Problems | .67 (.17–1.17) | .83 (.33–2.00) | 1.00 (.50–2.00) | 1.00 (.67–1.83) | KW(3) = 18.00; p<0.001 |
| Interpersonal Sensitivity | .50 (.00–1.00) | 1.00 (.25–1.75) | 1.25 (.50–2.19) | 1.50 (.50–2.25) | KW(3) = 42.25; p<0.001 |
| Depression | .33 (.00–1.00) | 1.00 (.33–2.00) | 1.17 (.33–2.29) | 1.50 (.67–1.83) | KW(3) = 46.45; p<0.001 |
| Anxiety | .33(.00-.83) | .67 (.33–1.33) | 1.00 (.21–1.96) | .83 (.33–2.00) | KW(3) = 23.76; p<0.001 |
| Hostility | .40 (.20-.60) | .60 (.20–1.00) | .40 (.20–1.20) | .80 (.20–1.40) | KW(3) = 14.81; p = 0.002 |
| Phobic Fear | .00 (.00-.40) | .20 (.00-.75) | .20 (.00–1.20) | .40 (.00–1.20) | KW(3) = 32.63; p<0.001 |
| Paranoid Ideation | .40 (.00–1.00) | .80 (.20–1.60) | .70 (.25–1.60) | 1.20 (.60–2.20) | KW(3) = 40.40; p<0.001 |
| Psychoticism | .20 (.00-.80) | .60 (.20–1.40) | .80 (.25–1.75) | 1.00 (.40–1.60) | KW(3) = 33.48; p<0.001 |
| Total score | .40 (.13-.89) | .81 (.36–1.60) | .98 (.38–1.56) | 1.04 (.49–1.62) | KW(3) = 42.93; p<0.001 |
| BSI-caseness (N;%) | 59 (28.0%) | 51 (49.5%) | 32 (53.3%) | 18 (51.4%) | Chi2(3) = 23.02; p<0.001 |
| **PANAS (mean; sd)** ** | | | | | |
| Positive Affect | 33.9 (6.7) | 29.2 (7.9) | 26.5 (6.6) | 22.6 (5.6) | F(3;154) = 14.66; p<0.001 |
| Negative Affect | 22.7 (9.0) | 28.5 (9.9) | 29.3 (9.5) | 30.4 (11.2) | F(3;154) = 6.07; p = 0.001 |
| **DAPP-sf (median; IQR)** ** | | | | | |
| Identity Problems | 9.0 (7.0–15.0) | 18.0 (11.0–23.5) | 20.0 (11.5–22.8) | 21.5 (15.3–25.3) | KW(3) = 32.32; p<0.001 |
| Submissiveness | 17.0 (12.0–23.5) | 23.0 (14.5–28.5) | 23.5 (18.0–31.0) | 18.0 (12.0–23.0) | KW(3) = 15.09; p = 0.002 |
| Cognitive Distortion | 8.0 (6.0–12.0) | 10.0 (8.0–19.0) | 12.0 (7.3–19.0) | 13.0 (9.3–14.0) | KW(3) = 11.16; p = 0.011 |
| Affective Instability | 18.0 (12.0–24.5) | 26.0 (18.0–31.0) | 30.0 (15.3–35.0) | 29.0 (18.5–33.0) | KW(3) = 21.91; p<0.001 |
| Stimulus Seeking | 16.0 (12.0–20.0) | 17.0 (12.5–23.5) | 17.0 (13.3–22.8) | 16.0 (10.0–20.0) | KW(3) = 1.80; p = 0.616 |
| Compulsivity | 18.0 (14.0–24.0) | 20.0 (16.0–26.0) | 22.0 (16.8–27.3) | 25.5 (15.8–29.5) | KW(3) = 6.89; p = 0.075 |
| Restricted Expression | 17.0 (12.0–25.0) | 22.0 (18.5–29.0) | 23.0 (20.3–27.0) | 28.5 (21.5–34.3) | KW(3) = 23.81; p<0.001 |
| Callousness | 17.0 (14.0–21.0) | 18.0 (15.0–22.0) | 15.0 (12.3–21.0) | 17.0 (12.3–21.5) | KW(3) = 3.33; p = 0.344 |
| Rejection | 22.0 (16.0–27.5) | 26.0 (20.0–33.5) | 26.0 (18.0–33.8) | 28 (18.3–21.3) | KW(3) = 6.362; p = 0.095 |
| Intimacy Problems | 16.0 (12.3–19.0) | 18.0 (14.5–22.5) | 17.5 (14.0–22.3) | 21.0 (14.0–29.0) | KW(3) = 8.927; p = 0.030 |
| Oppositionality | 18.0 (15.0–23.0) | 19.0 (13.0–24.5) | 16.5 (12.3–23.5) | 17.0 (12.3–21.3) | KW(3) = 1.42; p = 0.701 |
| Anxiousness | 13.0 (9.0–19.5) | 19.0 (12.0–24.5) | 23.5 (20.3–26.0) | 20.5 (13.3–27.8) | KW(3) = 24.43; p<0.001 |
| Conduct Problems | 9.0 (8.0–11.0) | 10.0 (8.0–13.5) | 10.0 (8.0–13.3) | 11.0 (9.0–13.8) | KW(3) = 12.04; p = 0.007 |
| Suspiciousness | 10.0 (8.0–15.0) | 15.0 (9.5–21.5) | 11.0 (9.0–13.8) | 20.0 (12.8–26.8) | KW(3) = 22.72; p<0.001 |
| Social Avoidance | 21.0 (16.5–23.5) | 15.0 (8.5–24.0) | 20.0 (11.5–22.8) | 16.0 (13.0–21.0) | KW(3) = 28.54; p<0.001 |
| Narcissism | 19.0 (14.0–23.0) | 23.0 (18.5–26.5) | 21.0 (14.3–27.0) | 25.5 (15.0–30.5) | KW(3) = 9.00; p = 0.046 |
| Insecure Attachment | 11.0 (7.0–17.5) | 15.0 (9.5–20.0) | 14.0 (11.0–18.8) | 18.5 (13.0–23.0) | KW(3) = 12.42; p = 0.006 |
| Self-Harm | 6.0 (6.0–6.0) | 7.0 (6.0–14.5) | 8.5 (6.0–13.0) | 9.5 (6.0–16.8) | KW(3) = 22.47; p<0.001 |
| DAPPsf-caseness (N;%) | 9 (11.1%) | 20 (48.8%) | 15 (62.5%) | 7 (58.3%) | Chi2(3) = 35.47; p<0.001 |
| **IIP-64 (median; IQR)** ** | | | | | |
| Domineering | 3.00 (1.00–7.25) | 6.00 (1.50–10.50) | 6.50 (2.25–11.00) | 6.00 (2.25–8.50) | KW(3) = 8.61; p = 0.035 |
| Vindictive | 2.50 (1.00–7.00) | 6.00 (2.50–11.00) | 8.00 (4.25–10.00) | 7.50 (4.00–12.00) | KW(3) = 19.21; p<0.001 |

*(Continued)*

**Table 4.** (*Continued*)

|  | Overall Secure | IF | IM | IFM | Test |
|---|---|---|---|---|---|
| Cold /Distant | 2.00 (.00–5.00) | 7.00 (2.50–12.00) | 10.00 (5.00–16.00) | 11.50 (6.00–17.00) | KW(3) = 34.28; p<0.001 |
| Socially Inhabitant | 4.00 (1.75–8.00) | 11.00 (3.00–18.00) | 13.50 (5.00–19.75) | 12.50 (5.75–17.25) | KW(3) = 27.49; p<0.001 |
| Non-assertive | 8.00 (3.00–13.25) | 11.00 (5.50–21.00) | 15.00 (11.50–21.00) | 16.50 (5.75–23.50) | KW(3) = 18.88; p<0.001 |
| Overly Accomodative | 7.00 (4.00–13.00) | 12.00 (5.00–17.50) | 15.50 (7.75–19.75) | 17.00 (15.00–20.50) | KW(3) = 18.02; p<0.001 |
| Self Sacrificing | 9.00 (3.00–15.00) | 14.00 (5.50–19.00) | 16.50 (8.25–19.00) | 18.50 (16.25–22.50) | KW(3) = 17.74; p<0.001 |
| Intrusive/ Needy | 5.00 (3.00–10.25) | 7.00 (4.00–12.00) | 7.50 (5.25–15.00) | 10.00 (6.50–12.00) | KW(3) = 7.86; p = 0.049 |
| **CTQ (median; IQR)** ** |  |  |  |  |  |
| Emotional Abuse | 7.0 (5.0–9.0) | 10.0 (7.0–13.0) | 11.0 (9.0–14.0) | 12.0 (10.3–16.0) | KW(3) = 31.53; p<0.001 |
| Physical Abuse | 5.0 (5.0–5.0) | 5.0 (5.0–7.0) | 5.0 (5.0–6.8) | 6.5 (5.0–9.8) | KW(3) = 6.47; p = 0.091 |
| Sexual Abuse | 5.0 (5.0–5.0) | 5.0 (5.0–6.0) | 5.0 (5.0–7.5) | 6.0 (5.0–9.8) | KW(3) = 10.69; p = 0.014 |
| Emotional Neglect | 11.0 (9.0–13.0) | 14.0 (12.0–17.0) | 15.0 (12.3–17.8) | 18.5 (17.0–21.8) | KW(3) = 53.37; p<0.001 |
| Physical Neglect | 5.0 (5.0–7.0) | 7.0 (5.0–10.0) | 7.0 (6.0–10.0) | 9.5 (7.3–11.0) | KW(3) = 31.19; p<0.001 |

* BSI Dutch and English samples compiled

** Dutch sample only.

BSI = Brief Symptom Inventory; CTQ = Childhood Trauma Questionnaire; DAPPsf = Dimensional Assessment of Personality Pathology short form; IIP-64 = Inventory of Interpersonal Problems with 64 items; PANAS = Positive and Negative Affect Scale; Overall Secure: a secure attachment style regarding mother, father and romantic partner; IF: secure attachment style regarding mother and romantic partner, but not father; IM: secure attachment style regarding father and romantic partner, but not mother; IFM: insecure attachment style towards father as well as mother, but not partner if available.

ANQ-sort to the ECR-r model results in a significant 0.07–0.10 point improvement of the goodness-of-fit measures. Thus the combination of ECR-r and ANQ-sort predict DAPP-sf caseness best.

## Discussion

Consistent with the suggestions of several attachment theorists and researchers [3,27,36–39] Fonagy and colleagues [12] developed the ANQ as a self-report instrument to assess current adult attachment styles dimensionally and on a relationship-specific basis using a Q-sort methodology.

In the construction of the ANQ the distinction between items that theoretically were considered attachment items versus those considered non-attachment items, was assumed to be important. In this study among English and Dutch respondents post-hoc analyses showed that the correlations between attachment and the non-attachment items were large. However, in the full data set we were able to find only one person who reported marginally higher scores on the non-attachment items than the attachment items. This finding suggests that participants do not make an explicit distinction between attachment and non-attachment statements, but that they value attachment related characteristics over non-specific characteristics when appraising a relationship.oO Obviously, positive or negative valences of a specific relationship might either buffer insecurity or exacerbate psychopathology [77], but we assume that, from a psychotherapeutic perspective, the attachment profiles are more relevant because of their associated internal working models with their enduring effects on other relationships as well [2,35]. Of course, this assumption needs further empirical scrutiny. For these reasons, we

**Table 5. ECR-r and ANQ-sort predicting DAPPsf- and BSI-caseness.**

| | Chi2 | df | p-value | 2Loglikelihood | Cox & Snell R$^2$ | Nagelkerke R$^2$ |
|---|---|---|---|---|---|---|
| **DAPP-sf caseness** | | | | | | |
| *Univariate* | | | | | | |
| ANQ profiles | 37.368 | 3 | <0.001 | 161.380 | 0.211 | 0.294 |
| *Stepwise* | | | | | | |
| *Step 1*: ECR-r subscales | 50.433 | 2 | <0.001 | 148.515 | 0.273 | 0.382 |
| *Step 2*: ANQ profiles | | | | | | |
| Total model | 66.263 | 5 | <0.001 | 132.486 | 0.343 | 0.479 |
| Change | 15.830 | 3 | 0.001 | | 0.070 | 0.097 |
| **BSI-caseness** | | | | | | |
| *Univariate* | | | | | | |
| ANQ profiles | 24.237 | 3 | <0.001 | 522.340 | 0.058 | 0.078 |
| *Stepwise* | | | | | | |
| *Step 1*: ECR-r subscales | 43.146 | 2 | <0.001 | 153.236 | 0.243 | 0.338 |
| *Step 2*: ANQ profiles | | | | | | |
| Total model | 49.351 | 5 | <0.001 | 147.031 | 0.273 | 0.380 |
| Change | 6.205 | 3 | 0.120 | | 0.030 | 0.042 |

ANQ = ANQ-sort; BSI = Brief Symptom Inventory; DAPPsf = Dimensional Assessment of Personality Pathology short form; ECR-r = Experiences in Close Relationships -revised.

decided to leave out the non-attachment items in studying the psychometric properties of the ANQ.

As psychotherapy patients as well as healthy women and participants from the general population took part in this study, maximum variability of the attachment variables was ensured. After psychometric analyses, using CFA and multi-group CFA, the theoretical structure of attachment with three attachment styles (secure, dismissing, preoccupied) was supported. Dismissive and preoccupied attachment showed to be two separate, albeit dependent, concepts. The similarities found in the factor structures of the English and the Dutch samples, and over the mother and father as attachment figures also suggest that the observed factor solution is robust. Fit was less optimal regarding romantic partner as attachment figure. With this constraint, that asks for further study, the ANQ proved to be capable of assessing the different attachment styles.

Consistent with the Q-sort procedure, subsequent LCA of the attachment items showed that the ANQ was also able to distinguish subgroups of respondents with similar attachment profiles with regard to different potential attachment figures. Fifty-two percent of the participants had a secure attachment style towards their mother and father, as well as their romantic partner if available, the Overall Secure (OS-) group. Twenty-four percent of the participants had only an insecure attachment style towards their father, the Insecure Father (IF-) group. Fifteen percent of the participants had only an insecure attachment style towards their mother, the Insecure Mother (IM-) group. And 10% of the participants reported an insecure attachment style towards their father as well as their mother, the Insecure Father and Mother (IFM-) group. Although not directly comparable, as different instruments have different underlying assumptions and as different populations are involved, the prevalence of well over half of the

participants being Overall Secure corresponds quite well with prevalence estimates of participants with a secure attachment style in other studies [19,78,79].

These results indicate that, whereas some people may have a dominant or generalized attachment style (the participants with an OS- or IFM-profile), others have attachment styles that are relationship specific (the participants with an IF- or IM-profile). Importantly, the results also show that these different attachment-style profiles are not only relevant from a theoretical, but also from a clinical perspective. Specifically in contrast with the IFM-participants, the overall secure participants experienced relationships with their key-figures more positively, they reported the fewest psychiatric symptoms, they reported the most positive and the least negative affectivity, the lowest personality pathology, the fewest interpersonal problems, and the lowest levels of abuse and neglect. The participants with an IF- or IM-profile scored in between, suggesting that some protection might come from a more differentiated attachment style with possibly more nuanced internal working models that clinically come with a more flexible attachment style [2,35]. The data also indicate higher levels of pathology as well as more emotional abuse and neglect for the participants who are insecure with mother than those who are insecure with father. These results suggest that both parents are important in offering a secure base and a safe haven but that the mother has a more influential role in so far that mental health problems are more clearly reflected in reports of poor mother–offspring attachment relationships. It seems worthwhile to study if this pattern might be different in circumstances where fathers spent more time with their children and have a more central role in their upbringing.

In our study even participants within the IFM-group reported a secure attachment to their romantic partner. Accordingly, even these persons seem to be able to form a secure attachment with their romantic partner at least at the time of reporting. This is perhaps good news, but regarding the prevalence it is probably the outcome of a selection bias, too. We suppose that participants who are not able to experience a secure attachment to a romantic partner have more difficulties in maintaining a stable romantic partner relationship and therefore a majority of these participants are without a romantic partner relationship, hence providing no data. This might also explain the less optimal fit of the theoretical model regarding romantic partner which implies that the non-attachment items might have played a greater significance in Q-sorts in these instances. In our study we favored the parsimonious and theoretically sensible four-factor model above the six-factor model that seemed to have a somewhat better fit statistically but was more difficult to interpret and was less robust. However, by opting for the more parsimonious model we might have reduced the chance of finding differences between subscales and attachment patterns, which might have been found in larger datasets. The contribution of the avoidant and preoccupied styles towards insecure attachment, however, are comparable as represented in Fig 1, while associations of the different attachment style profiles showed the strongest associations with the ECR-r scale measuring 'attachment related anxiety'.

The ANQ is not alone in differentiating attachment relationships. For example, Lindberg & Thomas [80] developed the Attachment and Clinical Issues Questionnaire (ACIQ) that probes for attachment styles towards mother, father and romantic partner. Mallinckrodt, Gantt & Coble [81] constructed the Client Attachment to Therapist Scale (CATS) to assess the attachment style patients display toward their psychotherapist. Maunder and Hunter [82] developed a self-report questionnaire assessing the attachment style of the patient towards health care providers in general. And lastly, Fraley, Heffernan, Vicary & Brumbaugh [83] studied a shortened version of the ECR-r (ECR-RS) for suitability as an instrument to assess relationship specific attachment styles. The ANQ-sort differs from these instruments in providing for an open-ended range of key-figures. It also differs from other self-report questionnaires by the

random presentation of the items each time a new key-figure is selected, making response bias less probable. Finally, it differs from other self-report questionnaires in employing Q-sort methodology, making it a quasi-qualitative instrument that allows for profiling subgroups of patients with similar attachment styles towards selected potential attachment figures [30].

In this study, the ANQ turned out to be an instrument with good acceptability and a completion time of a mean 40 minutes for three potential attachment figures. Although the ANQ-sort is a more elaborate instrument than the ECR-r and it has only limited value in addition to the ECR-r in predicting personality pathology, we believe its Q-sort procedure with randomly presented items as well as its possibility to assess attachment style towards different key-figures makes it a useful, alternative instrument for those clinicians as well as clinical researchers who want to assess the spectrum of individuals' attachment styles across key relationships. In addition, in a clinical context the discussion of a particular sort by a client can be the basis of reflective exploration.

Although the results are promising, this study has some additional limitations that need to be mentioned. This is the first analysis of the ANQ and additional research is needed to replicate the factor structure and to assess the extent to which the attachment style profiles found in this study will be replicated in other populations. Another limitation is that the two samples are recruited separately and differently, restricted, for example, to females only in the Dutch sample.

Meanwhile, interested clinicians can assess the profile of their individual cases by the correspondence of their scores with the mean scores of the participants in the OS-, the IF-, the IM- and the IFM-groups in this study (see S4 Supplementary Table 3; in S1 File Scale scores per profile group and S5 Appendix Scoring Tool ANQ in S2 File). Furthermore, the instrument will be useful to further investigate the contrast between relationship-specific and general models of attachment [35,84,85]. In psychotherapy praxis the ANQ can also be used to assess the attachment style as it developes towards a clinician as for example Mallinckrodt & Jeong [86] and Taylor, Rietzschel, Danquah & Berry [87] found relevant for the development of a good working alliance.

## Supporting information

**S1 File.**
(DOCX)

**S2 File.**
(XLSX)

## Author Contributions

**Conceptualization:** Cornelis G. Kooiman, Jon G. Allen, Peter Fonagy.

**Data curation:** Cornelis G. Kooiman, Nicolas Lorenzini, Jurate Aleknaviciute, Peter Fonagy.

**Formal analysis:** Astrid M. Kamperman, Cornelis G. Kooiman, Nicolas Lorenzini, Jurate Aleknaviciute, Peter Fonagy.

**Funding acquisition:** Cornelis G. Kooiman.

**Investigation:** Astrid M. Kamperman, Cornelis G. Kooiman, Nicolas Lorenzini, Jurate Aleknaviciute, Jon G. Allen, Peter Fonagy.

**Methodology:** Astrid M. Kamperman, Cornelis G. Kooiman, Jurate Aleknaviciute, Jon G. Allen, Peter Fonagy.

**Project administration:** Cornelis G. Kooiman, Nicolas Lorenzini.

**Supervision:** Jon G. Allen, Peter Fonagy.

**Writing – original draft:** Astrid M. Kamperman, Cornelis G. Kooiman, Jurate Aleknaviciute.

**Writing – review & editing:** Astrid M. Kamperman, Cornelis G. Kooiman, Nicolas Lorenzini, Jon G. Allen, Peter Fonagy.

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
