## [Decision Letter · Decision Letter 0]

4 Mar 2020

PONE-D-19-35791

Using the Attachment Network Q-sort for profiling one's attachment style with
different attachment-figures

PLOS ONE

Dear Dr Kooiman,

Thank you for submitting your manuscript to PLOS ONE. After careful consideration, we
feel that it has merit but does not fully meet PLOS ONE’s publication criteria as it
currently stands. Therefore, we invite you to submit a revised version of the
manuscript that addresses the points raised during the review process.

We would appreciate receiving your revised manuscript by Apr 18 2020 11:59PM. When
you are ready to submit your revision, log on to https://www.editorialmanager.com/pone/ and select the 'Submissions
Needing Revision' folder to locate your manuscript file.

If you would like to make changes to your financial disclosure, please include your
updated statement in your cover letter.

To enhance the reproducibility of your results, we recommend that if applicable you
deposit your laboratory protocols in protocols.io, where a protocol can be assigned
its own identifier (DOI) such that it can be cited independently in the future. For
instructions see: http://journals.plos.org/plosone/s/submission-guidelines#loc-laboratory-protocols

We look forward to receiving your revised manuscript.

Kind regards,

Alexandra Kavushansky, PhD

Academic Editor

PLOS ONE

Journal Requirements:

2. We note that you have included the ANQ items in the S1 Appendix. As these are
previously published and under copyright, as we understand, please remove them from
the appendix and instead describe in the Methods section which items were used in
the current study in sufficient detail such that other researchers could replicate
the analyses.

3. We note your manuscript is layed out as landscape rather than portrait, can you
therefore please amend your layout to portrait.

Additional Editor Comments (if provided):

The rationale and hypotheses behind testing two various sets of participants (English
vs. Dutch, psychiatric outpatients vs. controls) is not clearly stated neither in
the abstract, nor in the Introduction section. In the abstract there is a statement
that patients and non-patients were studied, without an explanation which patient
were checked (heart disease? stroke?) and how and why they were compared to
controls. Using DSM-IV criteria and axes is not justified either. Maybe I missed it,
looks like in the “Statistics” section not all the tests performed (as appears from
the Results section, e.g. tests comparing the 3 groups of patients, controls and
healthy participants) are listed. A stronger link between the Introduction section,
especially the hypotheses part, and the Discussion should be made.

Reviewers' comments:

Reviewer's Responses to Questions

**Comments to the Author**

1. Is the manuscript technically sound, and do the data support the conclusions?

Reviewer #1: Partly

2. Has the statistical analysis been performed
appropriately and rigorously? 

Reviewer #1: Yes

3. Have the authors made all data underlying the
findings in their manuscript fully available?

Reviewer #1: Yes

4. Is the manuscript presented in an intelligible
fashion and written in standard English?

Reviewer #1: Yes

5. Review Comments to the Author

Reviewer #1: This manuscript introduces a new measure (Attachment Network Q-Sort)
designed to assess relationship-specific attachment styles, and presents evidence to
support the validity of this measure. Overall the paper is well-written and I
believe has the potential to make a contribution to the literature on adult
attachment. Nonetheless, I have some lingering questions and concerns that could
stand to be addressed. I discuss these issues in the space below.

Introduction

• Although I think this measure has considerable promise, I got lost a bit in the
numerous explanations for its necessity. At various points the authors argue that
such a measure is a) necessary to circumvent length/participant burden (as with the
AAI), b) a matter of construct validity (pp. 6-7) that allows one to distinguish
between attachment and non-attachment related items, and c) essential for trying to
assess relationship-specific attachment styles. There is value in each of these
goals, but the flow of the introduction sometimes made it hard for me to understand
the value added of this particular measure, or what its main contribution to the
literature might be.

• In my view, the latter issue (assessing relationship-specific attachment styles) is
most compelling. That is, attachment research has historically done a relatively
poor job of accounting for the fact that many children are raised in the context of
multiple attachment figures and many adults maintain multiple attachment-relevant
relationships. To me this is the strongest case for a new measure in an already
crowded field of measures, and I’d like to see this justification emphasized more
strongly.

• In contrast, I struggled a bit to fully understand the case that this measure (more
than others) allows one to disentangle attachment with affective valence. That
argument was unclear to me from the beginning (though I grasp the general idea), and
I think perhaps there is a missed opportunity to establish more clearly discriminant
validity. i.e., to show that associations with other measures in this manuscript
differed among attachment vs. non-attachment items on the ANQ.

• There could be more clarity on the scoring of the ANQ. One can somewhat deduce this
by looking at the supplementary material, but there is in my view a lack of detail
in the text. It was not until the results section that I realized this produces
independent scores for secure, preoccupied, and dismissing styles. It is not
discussed explicitly how these are scored, or at least that is not obvious to me.
Especially given that other Q-sort based attachment measures (i.e, AQS) are
calculated by correlating with a “criterion sort” (which I presume was not done
here) I think it is important to be very clear in the explanation of scoring for
this particular instrument.

Method

• One critical issue that I was not clear on is the extent to which the ANQ for
mother/father is assessing representations of current relationships (as in the ECR)
or early caregiving experiences (as in the AAI). The decision to include the ECR and
not AAI as a measure of validity suggests that this is about current relationship
with parents, and the items themselves seem to indicate refer to current rather than
past relationship functioning. But this should be made abundantly clear throughout
the manuscript.

• This issue is also relevant when considering the length of the ANQ administration.
40 minutes is a relatively time-intensive instrument. It would represent a more
efficient alternative to the AAI, but it’s not clear that this is really designed to
be an alternative to the AAI. It is certainly much more time-consuming than adult
attachment measures such as the ECR. But I’m not entirely clear on whether it’s
meant to serve as a replacement for the ECR or a complement to it. If the former
then further justification would be needed for adding a much more time-intensive
instrument to the literature. Regardless, I think further clarification is
needed.

Results

• Overall, the results section seemed appropriate and generally clear. But in places
I was overwhelmed by the many different results reported. If it was possible to
condense and remove some redundant and/or unnecessary detail that might make this
section easier to follow.

• For instance, it did not seem to me that the paper (either in the introduction or
results/discussion) was particularly interested in distinctions between preoccupied
vs. dismissing classifications. I wonder if simply using the secure/insecure
distinction for the latent class analyses would be a simpler and cleaner way of
presenting these data.

• Perhaps more importantly is it necessary to present associations with every single
subscale of every instrument (BSI, DAPP, CTQ, etc.)? I understand the value
(particularly in a measurement validation paper) of having this information, but to
me the story gets lost or is made more difficult to follow when the reader has to
sort through many dozens of findings when a similar set of analyses using
higher-order composites might tell a similar story.

Discussion

• In this section I continued to be confused by the discussion of attachment vs.
non-attachment items (p. 32). I’m not suggesting that this isn’t an important issue.
Only that it’s not obvious to me what this means (for scientific and/or clinical
purposes) and this argument could perhaps be written more clearly.

• I think the manuscript could benefit from further discussion of generalizability of
this measure. Both in terms of which populations could use it, but also which
relationships could be asked about. Clarification on the extent to which ANQ
questions are generalizable to many types of relationships (beyond romantic partners
and parents) might be important for guiding future research in this domain.

6. PLOS authors have the option to publish the peer
review history of their article (what does this mean?). If published, this will
include your full peer review and any attached files.

If you choose “no”, your identity will remain anonymous but your review may still be
made public.

**Do you want your identity to be public for this peer review?** For
information about this choice, including consent withdrawal, please see our
Privacy Policy.

Reviewer #1: No

---

## [Author Response · Author response to Decision Letter 0]

16 Jun 2020

Editor #1

We checked the PLOS ONE’s style requirements and prepared the manuscript
accordingly.

Editor #2

The ANQ items are included in Appendix S1. As these are not under copyright, we
preferred to leave Appendix S1 as it is for clarity and readability of the
manuscript.

Editor #3

Accidently, initially the manuscript was submitted in landscape format. The layout is
now corrected in portrait format.

Editor #4

Unfortunately, ethical restrictions and patient confidentiality prohibit us from
making the databases publicly accessible. All data are stored at the institutional
database of the Erasmus Medical Centre, and will be made available upon request. At
the end of the methods section we inform the interested reader how to get access to
the files. Additionally, we have provided a maximum of information in our
supplementary material.

Editor additional comments:

We re-drafted the Discussion to make it more congruent with the hypothesis part of
the Introduction.

• In the Abstract we added that the study is done in psychotherapy patients as well
as in non-patient respondents.

• In the Methods section and in the Discussion, we explained that the study was done
in two samples (an English convenient sample from the general population and a Dutch
sample of psychotherapy patients and of healthy respondents) taken together to
ensure generisability of the instrument and the structures yielded by the study.

• In the Methods section we explicated that in the Dutch sample DSM-IV criteria were
applied necessary for the in- and exclusion criteria in the larger study of which
this psychometric study was a part.

• The statistics paragraph is checked for completeness. The tests the editor is
referring to are described as follows: ‘Association between the obtained attachment
subgroups and proximal variables were formally tested using Chi2-tests for
categorical variables, ANOVA’s for normally distributed continuous variables, and
Kruskal-Wallis tests for non-normally distributed continuous data. ’

Reviewer Introduction bullet 1 and 2

The reviewer is completely right in assuming that the most important quality of the
ANQ is its focus on assessing relationship-specific attachment styles. We now
emphasize this more strongly in the Discussion.

Reviewer Introduction bullet 3.

In the construction of the ANQ the distinction between items that theoretically were
considered attachment items versus non-attachment items, was assumed to be
important. However, this distinction is empirically not confirmed as both sets of
items are highly correlated with each other. As shown in table 3, and accompanying
text in the Results section discriminant validity of the ANQ results regarding the
non-attachment items was not shown This is now more clearly explained in the
Discussion.

Reviewer Introduction bullet 4

A description of the scoring of the ANQ is added in the methods section.

Reviewer Method bullet 1

We now explain throughout the manuscript that the ANQ assesses current attachment
styles towards the key-figures chosen.

Reviewer Method bullet 2

In the Discussion we added our reasons why the more elaborate ANQ could be an
alternative for the ECR(-r) which indeed is shorter.

Reviewer Results bullet 1

In the Results section we removed a detailed description of the results on the DAPPsf
to improve the readability of the manuscript. The details themselves are still
presented in the table.

Reviewer Results bullet 2

The reviewer is correct. Preoccupied and dismissive subscale scores point generally
in the same direction. However, the confirmative factor analysis showed that
combining the dismissive and preoccupied subscales into a single insecure subscale
did not result in a better fit (Model C in S2 tables 1a-c, see also model
interpretation at the end of S2). This statistical argument, in combination with the
underlying theory of ANQ which distinguished a preoccupied and dismissive component,
resulted in our decision to retain both subscales in the latent class analyses. It
might be possible that a larger dataset might have resulted in more classes, both in
terms of differentiation between the preoccupied and dismissive subscale, as well as
differentiation between attachment figures. 

Reviewer Results bullet 3

We understand the comment of the reviewer on the detailed information given. We tried
to shorten the text taking this into consideration in the Results and Discussion
sections of the manuscript but to retain the analytic information in the tables. If
the editor prefers, we are willing to remove this table from the text and move it to
the appendix.

Reviewer Discussion bullet 1

We understand that the discussion of the supposed importance of attachment versus
non-attachment items could be much better. We tried to improve this in the Results
and Discussion sections.

Reviewer Discussion bullet 2

We added in the Discussion possible applications of the ANQ.

to reviewers.docx
---

## [Editor Report · Decision Letter 1]

30 Jul 2020

Using the Attachment Network Q-sort for profiling one's attachment style with
different attachment-figures

PONE-D-19-35791R1

Dear Dr. Kooiman,

We’re pleased to inform you that your manuscript has been judged scientifically
suitable for publication and will be formally accepted for publication once it meets
all outstanding technical requirements.

Kind regards,

Alexandra Kavushansky, PhD

Academic Editor

PLOS ONE
---

## [Editor Report · Acceptance letter]

5 Aug 2020

PONE-D-19-35791R1 

Using the Attachment Network Q-sort for profiling one's attachment style with
different attachment-figures 

Dear Dr. Kooiman:

I'm pleased to inform you that your manuscript has been deemed suitable for
publication in PLOS ONE. Congratulations! Your manuscript is now with our production
department. 

Kind regards, 

on behalf of

Dr. Alexandra Kavushansky 

Academic Editor

PLOS ONE